# Sparsify the Weights but Let the Gradients Flow!

## Abstract

Sparsity has become one of the promising methods to compress and accelerate Deep Neural Networks (DNNs). Structured sparsity has garnered significant interest as a result of relatively modest hardware overhead and improved efficiency on contemporary DNN accelerators. In particular, N:M sparsity is attractive because of hardware accelerator architectures capable of harnessing specific variations of N:M structured sparsity, enhancing computational efficiency. Additionally, this form of sparsity holds considerable appeal for reducing the DNN memory footprint owing to their modest representation overhead. Although there have been efforts to develop training recipes for N:M structured sparsity, they primarily focus on low-sparsity regions ($\sim$50%). As a consequence, the performance of models trained using these approaches tends to decline when confronted with high-sparsity regions. In this work, we extensively study the effectiveness of existing training recipes for N:M structured sparsity at high-sparsity regions and argue that these methods fail to sustain the model quality on par with low-sparsity regions. We demonstrate that the significant factor contributing to this disparity is the presence of elevated levels of induced noise in the gradient magnitudes. In order to mitigate this undesirable effect, we present two new sparse training recipes, namely *"Mask Decay Gradient Flow (*MDGF*)"* and *"Structure Decay Gradient Flow (*SDGF*)"* which employ decay mechanisms to progressively restrict the flow of gradients. Our results demonstrate that enabling the propagation of gradients plays a crucial role in preserving superior model performance while simultaneously attaining a high level of sparsity. Our evaluations of diverse sparsity configurations demonstrate that the proposed methods consistently achieve SOTA accuracy against conventional sparse recipes in a range of attention-based models used for various tasks encompassing both vision (up to $\Delta(Acc) \sim +2\%$) and language (up to $\Delta(Acc) \sim +5\%$).

## 1 Introduction

Deep Neural Networks (DNNs) have achieved notable success in many domains, such as computer vision, language understanding, and machine translation. A prevailing tendency in state-of-the-art DNN models is the rapid increase in their model size over time. For example, T5 from Google (Raffel et al., 2019), OPT from Meta (Zhang et al., 2022a), and GPT-4 from OpenAI (OpenAI, 2023) have over 100 billion parameters. The exponential increase in model size poses significant obstacles to the deployment of these models, particularly in devices with limited computational resources. To address this hurdle, an expanding body of research proposes model compression techniques such as quantization (Shen et al., 2020; Kim et al., 2021; Zafrir et al., 2019; Zhang et al., 2020), sparsification (Evci et al., 2019; Guo et al., 2016; He et al., 2017; Yao et al., 2019), and distillation (Gou et al., 2021).

This paper focuses on sparsification, which involves selectively eliminating model parameters by imposing zero values upon them. The benefits of sparsification are multi-fold. Firstly, sparsification offers the potential to decrease computational requirements by avoiding multiplications involving pruned weights. Secondly, it reduces the memory usage by employing compressed sparse representations (Qin et al., 2021), unlocking the possibility of deploying large models in resource-limited devices (Seshadri et al., 2022). Lastly, sparsification saves energy by eliminating unnecessary memory accesses for pruned weights and bypassing ineffectual computations.

While appealing, sparsification predominantly revolves around the inherent trade-offs between the quality of the model and compression ratio[1]. For example, some studies (Guo et al., 2016; Han et al., 2015b) have demonstrated promising results in achieving unstructured sparsity levels of around 90%-95% in image classification models while maintaining the quality of dense models. Similarly, the noticeable achievements of transformer-based models, primarily driven by their exponential growth in model size (Wei et al., 2022), have stimulated interest (Child et al., 2019; Beltagy et al., 2020; Kitaev et al., 2020) in exploring sparsification recipes for such models with high sparsity ratio. This serves as a significant incentive for the sparsification of attention-based models, as it enables the pruning of a substantial number of model parameters (>70%), resulting in a remarkable reduction in model size while maintaining an acceptable level of accuracy (Tay et al., 2022; Jaszczur et al., 2021). Despite its inherent ability to trim the memory footprint of large models, the realization of unstructured sparsity in hardware poses nontrivial challenges for acceleration. The irregularity in the sparsity pattern hinders the efficient execution of sparse models by natively dense accelerators such as GPUs and TPUs. The sparsity-induced models frequently exhibit comparable or inferior performance to their dense counterparts because of the additional intricacies involved in compression/decompression of model parameters (Nvidia, 2021a; Ma et al., 2021; Renda et al., 2020; Lin et al., 2021; Gamboa et al., 2020; Zhu et al., 2019).

In light of this objective, structured sparsity has gained significant popularity because of its hardware-friendly characteristics, with a focus on regulating sparsity patterns such as channel/filter sparsity (Li et al., 2016; Wen et al., 2016; He et al., 2017) or block sparsity (Ma et al., 2021; Pool & Yu, 2021; Mishra et al., 2021; Nvidia, 2021b; Zhou et al., 2021). For example, dense accelerators can bypass an entire channel computation without requiring any hardware modifications. The caveat, however, these form of sparsification generally entails a higher magnitude of quality loss. Yao et al. (2019); Kang (2019) found that employing fine-grained N:M structured sparsity, which keeps N out of consecutive M elements, can mitigate the degradation in quality. Moreover, the debut of 2:4 structured-sparse tensor core in GPU Ampere architecture (Nvidia, 2021a) has generated additional enthusiasm in developing efficient N:M training recipes. Although recent methods (Pool & Yu, 2021; Mishra et al., 2021; Nvidia, 2021b; Zhou et al., 2021) demonstrate acceptable quality, their main focus lies in addressing sparsity levels up to 2:8. These methods, however, are less effective when dealing with high sparsity regimes such as 1:16, 1:32, and higher. Our studies show that elevated levels of induced noise in the gradient magnitudes constitute a notable contributing factor to the diminished model quality observed in current training recipes. This phenomenon can be primarily attributed to either the absence (Johnson & Zhang, 2013; Wang et al., 2013) or perturbation of gradient flow of existing sparse training recipes. Building on the insights from this study, we introduce alternative training recipes that demonstrate substantial improvements in model quality relative to existing methods, particularly under higher sparsity ratios. In summary, this paper presents the following contributions:

- **The impact of gradient perturbations becomes increasingly evident at elevated levels of sparsity, leading to a deterioration in the quality of the model.** We present empirical evidence that SR-STE, a state-of-the-art N:M structured training recipe (Zhou et al., 2021), is less effective at high sparsity regions, $> 75\%$. We attribute the lower performance of SR-STE to the nontrivial perturbation of gradient magnitudes, especially as the sparsity ratio increases. This perturbation during the initial stages of training[2] adversely amplifies the variance of gradients, resulting in a diminished model quality.

- **Gradient flow is all you need.** In order to alleviate the adverse effects caused by noisy gradients, we introduce two decaying-based sparse training recipes tailored for N:M structured sparsity: (1) *Mask Decay Gradient Flow* (MDGF) and (2) *Structure Decay Gradient Flow* (SDGF). The fundamental principle underlying both methods involves *progressively* limiting the propagation of gradients for pruned weights, while allowing the gradients to freely flow at the early stages of training. More specifically, MDGF gradually decays the sparsity mask, either linearly or exponentially, instead of employing the conventional binary mask. In contrast, SDGF encompasses a collection of iterative pruning methods that progressively increases the sparsity ratio, while maintaining the overall N:M sparsity patterns. We compare the proposed methods against SR-STE across range of transformer models and under different sparsity configurations. Our results demonstrate that the decaying-based

---

[1]In this paper, we designate algorithmic-wise factors such as accuracy, recall, and precision as *model quality.*. Additionally, we denote model runtime/latency as *model performance*.

[2]Recent studies (not particularly for sparse models) (Lu et al., 2023; Johnson & Zhang, 2013) have shown that the early stage of training (critical region) is imperative in the quality of training recipes.

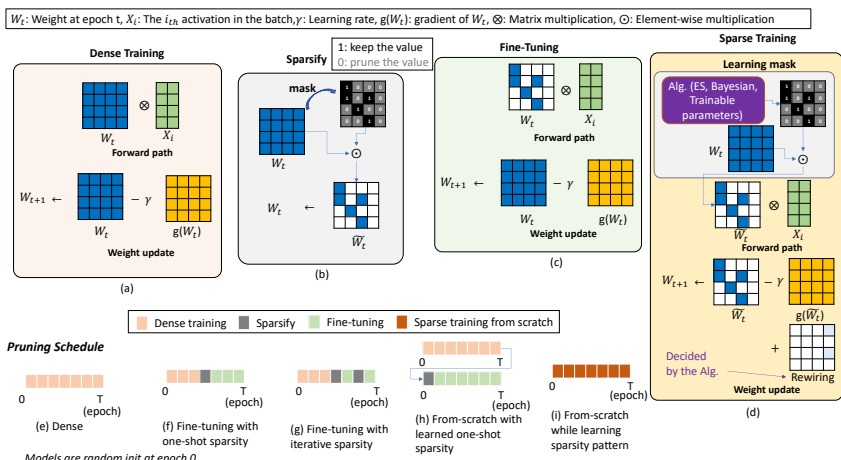

**Fig. 1: The computation flow of (a) Dense training, (b) Sparsify, (c) Fine-tuning, and (d) Sparse training. The training schedule of (e) regular dense training, (f) fine-tuning with one-shot sparsifying, (g) fine-tuning with iterative sparsifying, (h) from-scratch with learned one-shot sparsity pattern, and (i) from-scratch while learning sparsity pattern.**

methods consistently outperform SR-STE by up to 2%-5% in terms of model quality, while pruning ~97% of model parameters.

## 2 BACKGROUND AND RELATED WORKS

This work primarily focuses on weight sparsity, which poses a significant challenge in serving the ever-increasing model parameters of transformer-based models.

### 2.1 COMPUTATION FLOW OF SPARSE TRAINING RECIPES

Figure 1 summarizes the computation flows of various training recipes for the sparsification of model parameters. A sparsification recipe broadly entails 1) pruning criteria, 2) pruning schedule, and 3) sparsity pattern.

**(1) Pruning criteria.** The pruning criteria refers to the set of criteria used to determine the specific elements within the weight tensor that should be pruned. Magnitude pruning, which selects the pruning elements based on their absolute values, is one of the most widely used criteria for sparsification (Renda et al., 2020; Frankle & Carbin, 2019; Gale et al., 2019; Zhu & Gupta, 2017; Liu et al., 2018). Recent work employs other pruning criteria such as gradient (Yeom et al., 2021; Evci et al., 2020), Hessian (LeCun et al., 1989; Frantar & Alistarh, 2023), connection sensitivity (Lee et al., 2019), and importance estimation (Molchanov et al., 2019). In this paper, we employ magnitude pruning, following SR-STE (Zhou et al., 2021) the state-of-the-art structured N:M training recipe.

**(2) Pruning schedule.** We classify the pruning schedules into the following broad categories:

- *"Fine-tuning with one-shot pruning"* (Mishra et al., 2021; Pool & Yu, 2021; Frankle & Carbin, 2019; Lee et al., 2019), as shown in Figure 1f. This approach involves training a dense model, followed by 1-shot weight pruning , and finely the pruned model is retrained.
- *"Fine-tuning with iterative pruning"* (Evci et al., 2019; Han et al., 2015a; Guo et al., 2016; He et al., 2017; Molchanov et al., 2016; Yao et al., 2019; Zhu & Gupta, 2017; Gamboa et al., 2020; Narang et al., 2017a;b; Elsen et al., 2020; Evci et al., 2020), as shown in Figure 1g. This method trains a dense model followed by iterative cycles of pruning and re-training, which generally exhibits a greater capacity to regain lost quality.
- *"From-scratch with learned one-shot pruning pattern"* (Frankle et al., 2020; Evci et al., 2019), as shown in Figure 1h. This pruning schedule establishes the sparsity pattern based on the pretrained dense model and subsequently trains a sparse model from scratch.
- *"From-scratch while learning sparsity pattern"* (Wortsman et al., 2019; Dettmers & Zettlemoyer, 2019; Gale et al., 2019; Kusupati et al., 2020; Evci et al., 2020; Bellec et al., 2018; Mocanu et al.,

**Fig. 2: The weight update scheme of (a) SR-STE and (b)Methods proposed in this work.**

2018), as shown in Figure 1i. This method trains a sparse model from scratch while concurrently learning the sparsity pattern.

**(3) Sparsity pattern.** We broadly categorize sparsity patterns into following groups:

- *"Unstructured Sparsity"* refers to the process of pruning a model without imposing any constraints on the sparsity pattern (Renda et al., 2020; Guo et al., 2016; Lee et al., 2019; Frankle & Carbin, 2019; Gale et al., 2019). This can prune the model size to very small sizes while maintaing accuracy but often leads to increased runtime overhead.
- *"Coarse-grained Structured Sparsity"* enforces coarse-grained sparsity patterns, as its name implies, including techniques like filter/channel pruning (Li et al., 2016; Wen et al., 2016; He et al., 2017) and block-wise pruning (Wen et al., 2016; Ma et al., 2021; Narang et al., 2017b; Gray et al., 2017). This can achieve speedup in tranditional dense hardwares but results in a reduction in model quality.
- *"Fine-grained Structured N:M Sparsity"*, which prunes (M-N) out of M consecutive elements. Several preliminary studies rely on special threading and grouping techniques (Yao et al., 2019) or specialized sparse accelerators (Kang, 2019) to exploit this fine-grained sparsity pattern. With the inclusion of 2:4 structured-sparse GEMM support in tensor cores in GPU Ampere architecture (Nvidia, 2021a), recent work starts to investigate effective training recipes for N:M sparsity patterns to harness the existing accelerators (Pool & Yu, 2021; Mishra et al., 2021; Nvidia, 2021b; Zhou et al., 2021; Fang et al., 2022; Zhang et al., 2022b)

## 2.2 SPARSIFICATION OF ATTENTION MODELS

**Related Works.** Several related work has investigated N:M structured sparsity in attention-based models. SR-STE (Zhou et al., 2021) proposes a training recipe with fine-grained N:M structured sparsity from scratch. Figure 2(a) demonstrates the weight update scheme for the forward and backward pass of SR-STE. Nvidia ASP (Nvidia, 2021b) focuses on low sparsity (2:4) and employs channel permutations to maximize the accuracy of N:M sparse networks. SparseGPT (Frantar & Alistarh, 2023) introduces a post-training sparsification recipe tailored for GPT-family models. SparseGPT shows on-par model quality with up to 50% weight pruning under unstructured and N:M structured sparsity. LBC (Zhang et al., 2022b) learns best combinations during initial dense training, and than keeps the best combinations. IDP (Fang et al., 2022) iteratively prunes from dense to final target N:M ratio. No work focuses on the sparsifying attention model to N:M structured sparsity at a High sparsity ratio (>90%) .

## 3 DECAYING-BASED TRAINING RECIPES FOR SPARSITY

This section covers the proposed decaying-based training recipes for fine-grained N:M sparsity. These techniques are designed to gradually increase the sparsity ratio until the intended target sparsity level is attained. We classify the proposed decaying-based training recipes into two main categories, "**M**ask **D**ecay **G**radient **F**low" (MDGF) and "**S**tructure **D**ecay **G**radient **F**low" (SDGF), each with two variants which we discuss in details below. Unlike previous work (Zhou et al., 2021), we refrain from modifying the gradient update rules in either of these categories. Instead, we employ different gradual update rules for sparsity pattern or sparsity mask tensor, facilitating unimpeded gradient flow during the entire sparse training process.

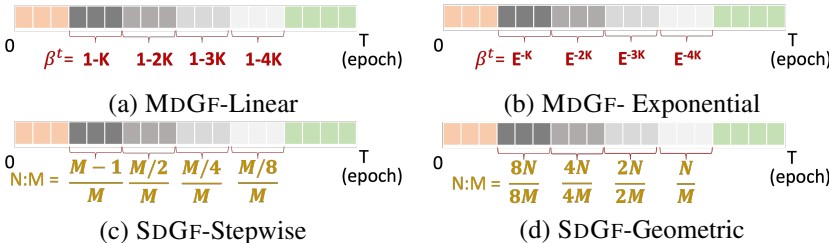

(a) MDGF-Linear      (b) MDGF- Exponential

(c) SDGF-Stepwise      (d) SDGF-Geometric

**Fig. 3: The sparsification schedules for MDGF and SDGF variants. The variables $\beta^t$ and N:M are same as explained in Figure 2(b).**

**Implementation.** In order to implement these methods, we employ the process of pruning dense weight tensors ($\mathcal{W}_t$) to generate sparse weight tensors ($\widetilde{\mathcal{W}}_t$), adhering to the following rule during the forward pass:

$$\widetilde{\mathcal{W}} = \mathcal{F}(\mathcal{W}, N, M, \Phi, \beta, j) = \mathcal{W} \circ [\Phi(\mathcal{W}, N, M, j) + \mathcal{D}(j)(1 - \Phi(\mathcal{W}, N, M, j))]$$

Here $\circ$ represents the Hadamard product between the matrices. $\Phi(\cdot)$ and $\mathcal{D}(\cdot)$ calculate a decaying-based binary mask and decay mask factor, respectively. Each function's implementations establish distinct decaying-based training recipes. $\Phi(\cdot)$ calculates a binary mask that matches the dimensions of the input weight tensor ($\mathcal{W}$). The location of 0s and 1s in the binary mask refers to pruned and unpruned weights, respectively. In fine-grained N:M structured sparsity with magnitude pruning, $\Phi(\cdot)$ assigns a value of 1 to the N weight tensor elements with the highest absolute magnitude within a contiguous block of M elements. Simultaneously, it enforces all the other elements within the block to be set to 0. In all of our experimental setups, we induce N:M sparsity along the row dimension of the weight tensor. In addition, $\mathcal{D}(\cdot)$ calculates the decaying factor for binary mask according to the target decaying-based training recipe.

❶ **Mask Decay Gradient Flow (MDGF).** In the first training recipe Figure 3 (a and b), we propose the use of a diminishing value ranging from 0 to 1, as opposed to the commonly-used binary pruning mask (e.g., "0" and "1" representing pruned and unpruned values, respectively). Note that for the mask-decay training recipes, the function $\Phi(\cdot)$ produces a mask tensor either with all ones (dense training) or with a sparsity pattern following target N:M fine-grained structured sparsity. In the initial phase of training, we use a mask comprising solely of ones and assign a constant value of 1 to $\mathcal{D}(\cdot)$, i.e., dense training.

Upon starting the sparse training phase, $\mathcal{D}(\cdot)$ produces gradually diminishing floating-point values between 1 and 0. The output of function $\mathcal{D}(\cdot)$ depends on the current decaying interval. Using a diminishing decaying factor facilitates unrestricted gradient flow for both pruned and unpruned weights. This is in contrast to prior work in which $\mathcal{D}(\cdot)$ is null which may cause instability in the training process. We propose two variations for $\mathcal{D}(\cdot)$, (a) *Linear* and (b) *Exponential*.

MDGF-*Linear* uses $\mathcal{D}(j) = max(1 - K_\tau \times j, 0)$ that reduces the decay mask values linearly with respect to training steps. In MDGF-*Exponential*, as its name implies, we use $\mathcal{D}(j) = e^{-K_\eta \times j}$, indicating an exponential decrease in the mask decay value relative to the ongoing training step. In both sparsity schedules, the value of $K_{\tau/\eta}$ determines the rate of decay. To ensure a binary mask value for the target N:M sparsity pattern, after sufficient decaying intervals, $\mathcal{D}(\cdot)$ approaches zero. After achieving the target N:M sparsity pattern, we proceed with few additional training epochs to restore the model accuracy. We postulate that the utilization of non-binary pruning mask values facilitates the smooth propagation of gradients in pruned weights, resulting in more stable sparse training and better model performance.

❷ **Structure Decay Gradient Flow (SDGF).** The second decaying-based training recipe decays the structure of the pruning mask, e.g. gradually altering the sparsity level, e.g. $\frac{3}{4} \mapsto \cdots \mapsto \frac{1}{4}$. In contrast to MDGF, this method strictly confines the pruning mask values to either 1 or 0, e.g. $\mathcal{D}(\cdot) = 0$. We propose two alternative implementations of $\Phi(\cdot)$, (a) *Stepwise* and (b) *Geometric*.

The SDGF-*Stepwise* starts by inducing M-1:M structured sparsity. Subsequently, it gradually increases the level of sparsity following $\frac{M}{2^d} : M$ formulation in which $d$ denotes the index of the decaying interval until $\frac{M}{2^d} == N$. For example, to retain a target sparsity level of 1:8, the method applies the following sparsity patterns at different decaying intervals $\frac{7}{8} \mapsto \frac{4}{8} \mapsto \frac{2}{8} \mapsto \frac{1}{8}$.

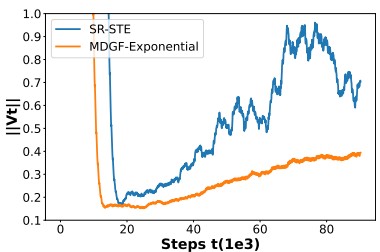

(a) The variance of the running avg. the second moment of the adamW gradient.

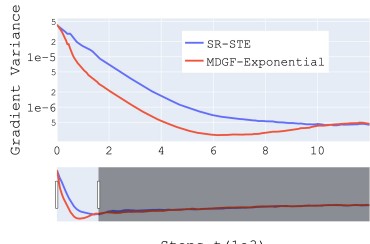

(b) The variance of gradient w.r.t. training steps.

**Fig. 4: Trends for different indicators during training. Data from ViT-tiny trained on CIFAR-10.**

The core idea of SDGF-*Geometric* is to maintain a constant ratio of $\frac{N}{M}$ throughout successive decay intervals by adjusting the values of N and M in proportion to each other. In all experiments, we configure $\Phi(\cdot)$ to be $\frac{k \times M}{2^d} : \frac{k \times N}{2^d}$. Unless specified otherwise, for our experiments, we start with $k = 16$. We empirically find that $k > 16$ offers negligible improvements in terms of model quality. For example, for a target sparsity of 1:8, we induce the following sparsity patterns at each decaying interval, $\frac{16}{128} \mapsto \frac{8}{64} \mapsto \frac{4}{32} \mapsto \frac{2}{16} \mapsto \frac{1}{8}$.

For both sparsity schedules, we evenly partition the total sparsification epochs throughout the decaying intervals. Figure 3 (c and d) illustrate the allocation of epochs for SDGF sparsity schedules. Fundamentally, this approach follows a hypothesis akin to MDGF. Enabling the flow of gradients of pruned weights throughout the model potentially leads to higher model accuracy.

## 4   WHY IS GRADIENT FLOW CRUCIAL TO THE PERFORMANCE OF SPARSIFICATION RECIPES?

To gauge the potency of the proposed sparsification methods before doing detailed studies, we will observe the training metadata. We will observe two key metrics. These metrics are widely used in the community for evaluating the efficacy of gradient-based learning. 1. Variance Change, $||V_t||$ studies indicate that lower variance of moments can help models learn better and converge faster. (Lu et al., 2023; Kingma & Ba, 2014) . 2. Gradient Variance: Lower variance in gradient will lead to faster convergence and better performance. (Johnson & Zhang, 2013; Wang et al., 2013)

We perform these studies on a Vision Transformer(ViT)-tiny with 3 encoder layers. We trained this model on the CIFAR-10 image classification dataset for 200 epochs with AdamW optimizer.

### 4.1   ASSESSMENT OF VARIANCE IN SECOND MOMENT ACROSS SPARSE TRAINING RECIPES

Figure 4(a) shows the variance of the second moment of gradient for Feed-Forward-2(FF2) in 1st layers of the model. During the training process, the variance steadily decreases in magnitude for MDGF, while in the case of SR-STE, the variance persists at a relatively high level even during the later stages of training. This suggests that higher noise in gradients for SR-STE, even during the later stages of training, would result in lower accuracy compared to the MDGF-exponential.

### 4.2   ASSESSMENT OF GRADIENT VARIANCE ACROSS SPARSE TRAINING RECIPES

Figure 4(b) shows the variance of absolute back-propagation gradients(which we use to quantify the noise in gradient). These are for gradients of the Feed-Forward-1(FF1) layer-0 in the model, [The trend is similar for all FF blocks in all layers as shown in the appendix figure, Figure 7]. For MDGF training, the variance of gradient falls quickly, in contrast, for SR-STE, the fall is gradual, taking a higher amount of steps. When the variance of the gradient is high, the optimizer spends time bouncing around, leading to slower convergence and worse performance. The variance for MDGF-Exponential comes down rather quickly thus the gradients are less noisy compared to SR-STE. This would result in higher accuracy for MDGF-Exponential.

These observations confirm that the proposed recipes can deliver better accuracy against SR-STE.

## 5 EXPERIMENT

In this section, we evaluate the effectiveness of various training recipes for N:M fine-grained structured sparsity in a range of attention-based models and tasks, encompassing image classification, language translation and understanding. Motivated by the relatively substantial contribution of FF layers (Section 11) in total FLOPs (∼64%) and parameter count (∼63.4%), we center our experiments around mainly sparsification of these layers within the encoder and decoder blocks. In addition, we conduct experiments on the pruning of projection layers ($\mathcal{Q}/\mathcal{K}/\mathcal{V}$) for a variant of `ViT-Base` (Dosovitskiy et al., 2021), a variant of `SwinV2-Base` (Liu et al., 2022), and `T5X-Base` (Raffel et al., 2019). For image classification tasks, we branched (commit: 1304589) our implementation from PyTorch Image Models (Wightman, 2019) and use NVIDIA A100 GPUs for training on ImageNet-1K dataset (Deng et al., 2009). For `T5X-Base`, we extend the official Google T5X release (commit: d3d3cbf) with sparsification training recipes and use Google TPUv3. We train these models from scratch using different training recipes across different patterns of N:M fine-grained structured sparsity. SR-STE serves as the baseline sparse training recipe to assess the effectiveness of the proposed training recipes in terms of model accuracy. Section 9 presents a detailed compilation of training hyperparameters, dataset details, and evaluation metrics.

### 5.1 IMAGE CLASSIFICATION ↦ `ViT-Base` AND `SwinV2` AND `ResNet50`

**Table 1: ImageNet-1K Top-1 validation accuracy on `ViT-Base` across different N:M sparsity patterns and training recipes.**

| Sparse Target | Dense | SR-STE | MDGF-Linear | MDGF-Exponential | SDGF-Stepwise | SDGF-Geometric |
|---|---|---|---|---|---|---|
| 2:4 (FF) | 76.389 | **77.761** | 77.613 | 76.381 | 77.081 | 77.363 |
| 1:4 (FF) | 76.389 | **78.782** | 78.512 | 78.579 | 77.357 | 78.347 |
| 1:8 (FF) | 76.389 | 77.869 | 78.019 | 78.009 | 77.025 | **78.175** |
| 1:16 (FF) | 76.389 | 75.637 | 76.594 | **77.325** | 75.923 | 76.869 |
| 1:32 (FF) | 76.389 | 73.056 | 75.807 | **76.068** | 74.394 | 74.910 |
| 1:128 (FF) | 76.389 | 72.069 | 74.012 | **74.180** | 71.725 | 69.801 |
| 1:4 (FF) + 1:4 (QK) | 76.389 | 78.145 | 77.755 | 78.113 | 77.163 | **78.229** |
| 1:8 (FF) + 1:8 (QK) | 76.389 | 75.527 | 76.473 | **77.349** | 76.617 | 76.334 |
| 1:8 (FF) + 1:4 (QK) | 76.389 | 78.144 | 78.025 | **78.273** | 77.163 | 76.839 |
| 1:8 (FF) + 1:4 (QKV) | 76.389 | 78.222 | 78.319 | **78.319** | 77.309 | 78.213 |

**`ViT-Base` model quality.** Table 1 presents Top-1 validation accuracy for variations of N:M sparsity in `ViT-Base`, with the highest accuracy model indicated in bold. The "*Sparse Target*" column signifies the intended level of N:M sparsity. For example, a sparsity target of 1:32 indicates that sparse tensors exhibit at most one non-zero for every 32 contiguous elements. In low sparsity scenarios (e.g., 2:4 and 1:4), both MDGF and SR-STE demonstrate comparable performance. However, with increases in sparsity degree (e.g., 1:8 and higher) employing SR-STE is detrimental to model quality. In contrast, the proposed decaying-based training recipes, MDGF-Exp and SDGF-Geo, yield the highest accuracy. Interestingly, when aiming for a sparsity target of 1:32 (approximately 97%), MDGF-Exponential showcases a mere 0.3% reduction in accuracy compared to a fully dense model (76.389 vs. 76.068). Additionally, we notice that the model accuracy increases at modest sparsity degrees, specifically in 2:4/1:4/1:8 (FF) patterns, resulting in an improvement of up to $\Delta(Acc) = +2.4\%$ in 1:4 (FF). The increase in model accuracy, can be attributed to Occam's Hill, wherein the positive impact of sparsity as a means of regularization is elucidated (Rasmussen & Ghahramani, 2001). In summary, the performance of MDGF-Exponential training recipe is comparable to that of SR-STE when dealing with low-sparsity scenarios. MDGF-Exponential performs best among all at high-sparsity regions.

**`SwinV2-Base` model quality.** Table 2 demonstrate Top-1 validation accuracy for `SwinV2-Base`. Similar to `ViT-Base`, we observe that the proposed decaying-based algorithms outperforms SR-STE across various N:M sparsity patterns. In 1:4 and 1:8 ($\mathcal{FF}$), SDGF-Geometric yields the highest Top-1 validation accuracy. Whereas, in the realm of high-sparsity patterns, MDGF-Exponential demonstrates superior performance compared to SR-STE. To summarize, the results from the two image classification models demonstrate that the proposed training recipes, MDGF and SDGF, which incorporate decaying-based approaches for N:M fine-grained structured sparsity, yield superior performance compared to SR-STE.

**Table 2: ImageNet-1K Top-1 validation accuracy on `SwinV2-Base` across different N:M sparse patterns and training recipes.**

| Sparse Target | Dense | SR-STE | MDGF-Exponential | SDGF-Stepwise | SDGF-Geometric |
|---|---|---|---|---|---|
| 1:4 (FF) | 83.45 | 82.355 | 82.491 | 82.267 | **82.469** |
| 1:8 (FF) | 83.45 | 81.437 | 81.382 | 81.382 | **81.466** |
| 1:16 (FF) | 83.45 | 80.154 | **80.542** | 80.386 | 80.274 |
| 1:32 (FF) | 83.45 | 78.972 | **79.545** | 76.480 | 79.277 |
| 1:8 (FF) + 1:8(QK) | 83.45 | 81.441 | **81.550** | 81.218 | 81.438 |

**Table 3: ResNet-50 Top-1 validation accuracy for different N:M sparsity patterns.**

| Sparse Target | Dense | SR-STE | MDGF-Exponential | SDGF-Stepwise |
|---|---|---|---|---|
| 2:8 | 85.09 | 83.33 | **83.60** | 82.97 |
| 1:8 | 85.09 | 80.78 | **82.48** | 81.17 |

**Table 4: The comparisons of the training recipes performance (FLOPs) and model accuracy.**

| Model | Sparsity recipe | Sparsity Target | Top-1 Acc (%) ↑ | Param (M) ↓ | FLOPs (G)↓ |
|---|---|---|---|---|---|
| ViT-Base | - | Dense | 76.389 | 85.70 (100%) | 33.29 (100%) |
| ViT-Base | SR-STE | 1:16 (FF) | 75.637 | 32.61 (38%) | 12.48 (37.5%) |
| ViT-Base | MDGF-Exp | 1:16 (FF) | 77.325 | 32.61 (38%) | 12.48 (37.5%) |
| ViT-Base | SR-STE | 1:8 (FF + KV) | 75.527 | 23.77 (27.7%) | 9.01 (27.08%) |
| ViT-Base | SDGF-Geometric | 1:8 (FF + KV) | 77.349 | 23.77 (27.7%) | 9.01 (27.08%) |

**ResNet-50 model quality.** While we focus majorly on sparsifiying attention based networks, for sake of completeness, we also test the efficacy of our recipe on CNNs. For this purpose, we train `ResNet-50` on Cifar-10, and sparsify all the convolution layers. We do this for SR-STE, MDGF-Exp, and SDGF-Stepwise. Table 3 shows the Top-1 validation accuracy. We see that similar to `ViT` and `Swin`, here also MDGF-Exponential outperforms SR-STE in both cases.

**Training performance.** Note that no off-the-shelf accelerator can natively support high-sparsity patterns. In order to assess the potential performance benefits of various training recipes, we employ an analytical cost model to estimate the savings in training and inference FLOPs as well as memory usage. Table 4 depicts that implementing a sparsity pattern of 1:16 ($\mathcal{FF}$), there is a reduction of 62% in the size of model parameters, along with a decrease of approximately 62.5% in the inference cost measured in FLOPs. MDGF-Exponential has 1.68% better accuracy than SR-STE at 1:16. For 1:8 sparsification of $\mathcal{FF}$ and $\mathcal{QK}$ weights, the model parameters and inference FLOPs are reduced by $\approx 73\%$. Section 11 details the numerical calculation for parameters and flops shown in Table 4. In this case, MDGF-Exponential has 1.82% better accuracy than SR-STE.

## 5.2 LANGUAGE UNDERSTANDING ↦ `T5X-Base`

Additionally, we analyze the effectiveness of the proposed decaying-based training recipes for the language understanding task. We employ a dense pre-trained `T5X-Base` model trained on the C4 dataset with a span-corruption objective (Raffel et al., 2019). The dense pre-trained model undergoes fine-tuning using the GLUE dataset (Wang et al., 2019) with various training recipes for N:M structured sparsity. Table 5 depicts the overall score, summarized across eight different GLUE tasks. We observe a consistent trend where SDGF outperforms SR-STE at high-sparsity patterns and an increasing number of sparse layers. Notably, we observe a relative difference of $\Delta = +5.3$ in 1:8 ($\mathcal{FF}$) + 1:8 ($\mathcal{QKV}$) sparsity pattern. Section 8.1 and Section 8.2 provide details about the `T5X-Base` model, per-task evaluation metrics, and additional ablation studies.

## 5.3 LANGUAGE TRANSLATION ↦ `Enc-Dec`

Next, we compare the performance of different sparse training recipes on WMT language translation task (Bojar et al., 2017). For that, we use an encoder-decoder transformer-based model (Vaswani et al., 2017). Section 9 outlines the details about this model and the training hyperparameters.

**Table 5: The GLUE overall score on the sparsified `T5X-Base` model across different N:M sparse training recipes and patterns.**

| Model | Sparsity Target | Dense | SR-STE | SDGF-Stepwise | SDGF-Geometric |
|---|---|---|---|---|---|
| T5X-Base | 1:4 (FF) | 86.2 | **84.1** | 83.7 ($\Delta = -0.4$) | 83.4 |
| T5X-Base | 1:32 (FF) | 86.2 | 79.4 | **80.9 ($\Delta = +1.5$)** | 79.3 |
| T5X-Base | 1:8 (FF) + 1:8 (QK) | 86.2 | 75.8 | **80.7 ($\Delta = +4.9$)** | 76.8 |
| T5X-Base | 1:8 (FF) + 1:4(QKV) | 86.2 | 78 | **80.3 ($\Delta = +2.3$)** | 78.9 |
| T5X-Base | 1:8 (FF) + 1:8 (QKV) | 86.2 | 74.2 | **79.5 ($\Delta = +5.3$)** | 75.8 |

Table 6 demonstrates the accuracy results across range of sparsity patterns and training recipes. We observe that SDGF and MDGF collectively outperform SR-STE across various N:M structured sparsity patterns. However, we note that the difference in accuracy achieved through different training recipes is relatively smaller. This can be attributed to the smaller model size, as well as the nature of the translation task, which appears to be less sensitive to sparsity patterns and training recipes[3].

**Table 6: The translation accuracy across different N:M sparsity patterns and training recipes.**

| Model | Sparsity Target | Dense | SR-STE | SDGF-Stepwise | MDGF-Exp |
|---|---|---|---|---|---|
| Enc-Dec (WMT) | 1:16 | 74.7 | 70.9 | **71.7** | **71.7** |
| Enc-Dec (WMT) | 1:32 | 74.7 | 70.7 | 71.3 | **71.4** |
| Enc-Dec (WMT) | 1:64 | 74.7 | 70.7 | 71.0 | **71.1** |
| Enc-Dec (WMT) | 1:128 | 74.7 | 70.7 | 70.8 | **71.1** |

## 5.4 BASELINE COMPARISONS

SR-STE is our primary baseline in our evaluations as it has shown good results in low-sparsity regions [2:4,1:4] and is considered SOTA for N:M training. We also compared against other techniques like Inherited Dynamic Pruning(IDP)(Fang et al., 2022), and SNIP: Single-shot Network Pruning(Lee et al., 2019). Table 7 compares the results on T5X with GLUE dataset. We also tried to test against LBC(Zhang et al., 2022b) but could not recreate the results shown in the paper.[4]

| Sparse Target | SR-STE | SNIP | IDP | MDGF-Exponential |
|---|---|---|---|---|
| 1:32 (FF) | 79.4 | 79.5 | 80.6 | **80.9** |

**Table 7: Comparing various sparsification techniques by fine-tuning T5X on GLUE dataset.**

## 6 LIMITATIONS AND FUTURE WORKS

The prevalence of self-attention models and their growing parameter size inspired this work to study the impact of sparsity for such models. In this work, we only study each sparsification recipe in isolation (either MDGF or SDGF). Nonetheless, combining these methods (SDGF-Geo + MDGF-Exp) at different training regions may lead to better training recipe for structured sparsity.

## 7 CONCLUSION

This work studies the efficacy of recent sparsity recipes for structured N:M sparsity across range of transformer-based DNN models. We observe that conventional methods introduce nontrivial noise to gradient values, especially at high-sparsity regimes (>75%). Building on this observation, we propose and compare two new decaying-based training recipes, namely MDGF and SDGF, for N:M structured sparsity. Our results demonstrate that our method, MDGF-Exponential and SDGF-Geometric consistently deliver SOTA model accuracy for a variety of vision and language models, with more than ~2% (Vision) and ~5% (Language) improvement at high structured sparsity regime compared to other pruning techniques.

---

[3]Model accuracy is less affected as we increase the sparsity level beyond 1:32.
[4]We have contacted the authors but cannot solve the issue.

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

## 8 APPENDIX

### 8.1 ABLATIONS STUDIES

This section shows the various ablation studies we performed during our experiments.

#### 8.1.1 EFFECT OF DENSE TRAINING STEPS ($d$)

Both our proposed methods, MDGF and SDGF include a dense training phase. We do an ablation study on different amounts of dense training steps(% of total steps) in Table 8. We perform this study on the language translation model (more implementation details in section §9.2.4) trained on WMT-17. We found that changing the dense step between 1.25% - 10% of the total training steps does not observably change the accuracy performance. However, empirically, we found that the dense training phase is still essential. The model cannot achieve as competitive accuracy without few epochs of dense training.

**Table 8: Ablation: The effect of number of dense training steps ($d$).**

| Accuracy | | MDGF-Linear | | | | SDGF-Stepwise | | | |
|---|---|---|---|---|---|---|---|---|---|
| Sparsity Target | | 1:16 | 1:32 | 1:64 | 1:128 | 1:16 | 1:32 | 1:64 | 1:128 |
| | 1.25% | 0.7155 | 0.7134 | 0.7106 | 0.7100 | 0.7157 | 0.7134 | 0.7108 | 0.7106 |
| Dense steps (d) | 2.5% | **0.7160** | 0.7127 | **0.7110** | 0.7093 | 0.7160 | 0.7136 | **0.7117** | 0.7100 |
| | 5% | 0.7157 | **0.7137** | 0.7103 | 0.7094 | 0.7164 | **0.7141** | 0.7107 | 0.7098 |
| | 10% | 0.7156 | 0.7126 | 0.7107 | **0.7104** | **0.7165** | 0.7128 | 0.7115 | **0.7107** |

#### 8.1.2 EFFECTS OF FINE-TUNING STEPS ($s$)

We also have a sets of study on number of fine-tuning steps in Table 9. We perform this study on the language translation model (more implementation details in section §9.2.4) trained on WMT-17. We found that for all of our proposed methods, the fine-tuning steps between 10% - 20% of the total training steps do not observably change the accuracy performance. However, empirically, we also found few steps of fine-tuning at the end are essential to recovering the accuracy.

**Table 9: Ablation: The effect of number of fine-tuning steps ($s$).**

| Accuracy | | MDGF-Linear | | | | SDGF-Stepwise | | | |
|---|---|---|---|---|---|---|---|---|---|
| Sparsity Target | | 1:16 | 1:32 | 1:64 | 1:128 | 1:16 | 1:32 | 1:64 | 1:128 |
| Fine-tuning steps (s) | 10% | 0.7153 | 0.7130 | **0.7107** | **0.7098** | **0.7160** | **0.7125** | **0.7095** | **0.7072** |
| | 20% | **0.7161** | **0.7132** | 0.7106 | 0.7097 | 0.7121 | 0.7093 | 0.7081 | 0.7065 |

#### 8.1.3 EFFECT OF ($\beta^t$) IN MDGF-LINEAR

We also study on effect of decay rate on model's accuracy in Table 10. We do experiments with varying $\beta^t$ for ViT-Base trained on Imagenet-1k for different sparsity targets.

We observe that a higher decay rate is beneficial at low sparsity targets (2:4,1:4), but for targets higher than 1:8, we found lower decay rate works better.

**Table 10: Ablation: The effect of mask decay rate ($\beta^t$) for MDGF-Linear.**

| Sparsity Target | | 2:4 | 1:4 | 1:8 |
|---|---|---|---|---|
| Mask decay rate ($\beta^t$) | 0.0002 | 77.495 | 78.448 | **78.019** |
| | 0.001 | **77.613** | **78.512** | 76.4075 |

### 8.2 DETAILED RESULTS FOR T5X-Base SPARSIFICATION ON GLUE DATASET

We compared sparsification methods N:M block sparsification against state-of-the-art technique, SR-STE on. T5 model uses a span-based masked language modeling (MLM) objective. T5 models were introduced in Raffel et al. (2019) and the updated models are available at T5X-github. We train a pre trained t5x-base model on GLUE dataset (Wang et al., 2019).

The main paper shows a snapshot of the performance across various sparsity targets using the overall score as metric. Table 11 presents all 9 scores for each sparsification technique and sparsity target.

**Table 11: GLUE full score using various T5X-base with different N:M sparse targets and various sparsification techniques.**

| | | overall score | CoLA | MNLI matched | MNLI mismatched | MRPC | QNLI | QQP | RTE | SST-2 | STS-B |
|---|---|---|---|---|---|---|---|---|---|---|---|
| Dense | - | 86.2 | 58.9 | 87.2 | 87 | 92.4 / 89.2 (90.8) | 93.6 | 92.0 / 89.2 (90.6) | 82.3 | 95 | 90.1 / 90.0 (90.0) |
| SR-STE (Zero Dense) | 1:4 | 83.1 | 41.8 | 85.2 | 85.3 | **92.8 / 90.0 (91.4)** | 92.3 | 91.8 / 88.9 (90.3) | 79.1 | **93.6** | **89.5 / 89.2 (89.3)** |
| SR-STE (10K Dense) | 1:4 | **84.1** | 48.1 | **85.7** | **85.6** | 92.4 / 89.5 (91.0) | 92.1 | **91.8 / 89.0 (90.4)** | **82.7** | **93.6** | 87.9 / 87.7 (87.8) |
| MdGf-Stepwise (10K Dense) | 1:4 | 83.7 | **48.8** | 85.3 | 85.4 | 92.4 / 89.2 (90.8) | 92.3 | 91.8 / 89.0 (90.4) | 80.5 | 93.5 | 86.5 / 86.3 (86.4) |
| MdGf-Geometric (Zero Dense) | 1:4 | 83.3 | 48.4 | 85.3 | 85.3 | 92.0 / 89.0 (90.5) | 91.8 | 91.8 / 88.9 (90.3) | 78 | 92.8 | 87.3 / 87.4 (87.3) |
| MdGf-Geometric (10K Dense) | 1:4 | 83.4 | 47.2 | 85.4 | 85.3 | 92.6 / 89.7 (91.1) | 92 | **91.8 / 89.0 (90.4)** | 79.8 | 92.9 | 86.7 / 86.4 (86.5) |
| SR-STE (Zero Dense) | 1:32 | 77.1 | 19 | 81.3 | 81.3 | 90.9 / 87.0 (89.0) | 86.9 | 90.6 / 87.4 (89.0) | 71.1 | 89.9 | 86.7 / 86.8 (86.8) |
| SR-STE (10K Dense) | 1:32 | 79.4 | 29.4 | 82.2 | 82.6 | 91.5 / 88.5 (90.0) | 89.6 | 91.2 / 88.2 (89.7) | 72.6 | **91.4** | **87.1 / 87.2 (87.2)** |
| MdGf-Stepwise (10K Dense) | 1:32 | **80.9** | **38.3** | **83.6** | **83.7** | **92.5 / 89.7 (91.1)** | **90.5** | **91.5 / 88.5 (90.0)** | **74.4** | 91.2 | 85.2 / 85.0 (85.1) |
| MdGf-Geometric (Zero Dense) | 1:32 | 77.6 | 20.2 | 81.3 | 81.6 | 91.8 / 88.5 (90.1) | 87.2 | 90.8 / 87.7 (89.2) | 73.3 | 90.1 | 85.8 / 85.5 (85.6) |
| MdGf-Geometric (10K Dense) | 1:32 | 79.3 | 29.2 | 82.3 | 82.9 | 91.3 / 88.0 (89.6) | 90.4 | 91.3 / 88.3 (89.8) | 73.3 | 90.5 | 85.4 / 85.4 (85.4) |
| SR-STE (Zero Dense) | 1:8(FF) + 1:8(QK) | 74.4 | 15.7 | 77.2 | 77.6 | 89.9 / 85.8 (87.8) | 83.6 | 89.7 / 86.2 (87.9) | 67.5 | 88.2 | 84.1 / 83.9 (84.0) |
| SR-STE (10K Dense) | 1:8(FF) + 1:8(QK) | 75.8 | 19.9 | 78.6 | 79.4 | 89.7 / 86.0 (87.9) | 84 | 90.1 / 86.7 (88.4) | 70 | 89.4 | 84.5 / 84.2 (84.4) |
| MdGf-Stepwise (10K Dense) | 1:8(FF) + 1:8(QK) | **80.7** | **38.7** | **83.1** | **83.2** | **90.9 / 87.7 (89.3)** | **89.9** | **91.2 / 88.2 (89.7)** | **76.2** | **91.9** | **84.5 / 84.5 (84.5)** |
| MdGf-Geometric (Zero Dense) | 1:8(FF) + 1:8(QK) | 75.8 | 21.6 | 78.8 | 79 | 90.0 / 86.0 (88.0) | 83.6 | 90.1 / 86.6 (88.3) | 69.7 | 88.9 | 84.0 / 83.9 (83.9) |
| MdGf-Geometric (10K Dense) | 1:8(FF) + 1:8(QK) | 76.8 | 22.3 | 80.7 | 80.9 | 89.8 / 85.8 (87.8) | 86.3 | 90.5 / 87.4 (89.0) | 70 | 91.1 | 83.7 / 83.4 (83.6) |
| SR-STE (Zero Dense) | 1:8(FF) + 1:8(QKV) | 73.2 | 13.5 | 76.3 | 76.4 | 89.0 / 84.6 (86.8) | 83.2 | 89.5 / 85.9 (87.7) | 63.9 | 87 | 84.3 / 84.2 (84.2) |
| SR-STE (10K Dense) | 1:8(FF) + 1:8(QKV) | 74.2 | 16.1 | 77.7 | 77.6 | 88.5 / 84.1 (86.3) | 82.9 | 89.9 / 86.3 (88.1) | 66.4 | 88.8 | 84.4 / 84.2 (84.3) |
| MdGf-Stepwise (10K Dense) | 1:8(FF) + 1:8(QKV) | **79.5** | **33** | **82.3** | **82.3** | **91.3 / 87.7 (89.5)** | **89.2** | **91.0 / 88.0 (89.5)** | **74.4** | **91.1** | **84.5 / 84.8 (84.6)** |
| MdGf-Geometric (Zero Dense) | 1:8(FF) + 1:8(QKV) | 75.5 | 22.1 | 78.6 | 78.7 | 90.5 / 86.8 (88.6) | 83.4 | 90.0 / 86.5 (88.2) | 67.9 | 88.2 | 84.2 / 84.2 (84.2) |
| MdGf-Geometric (10K Dense) | 1:8(FF) + 1:8(QKV) | 75.8 | 19.5 | 79.4 | 79.6 | 89.4 / 85.3 (87.3) | 84.5 | 90.2 / 86.8 (88.5) | 70.4 | 89.8 | 83.3 / 83.0 (83.2) |
| SR-STE (Zero Dense) | 1:8(FF) + 1:4(QKV) | 75.1 | 15 | 78.4 | 79 | 90.5 / 86.8 (88.6) | 84.2 | 90.1 / 86.6 (88.4) | 67.9 | 88.4 | **86.2 / 86.1 (86.2)** |
| SR-STE (10K Dense) | 1:8(FF) + 1:4(QKV) | 78 | 24.5 | 81.2 | 81.6 | 91.1 / 87.7 (89.4) | 87.1 | 90.6 / 87.3 (89.0) | 72.2 | 90.9 | 85.8 / 85.8 (85.8) |
| MdGf-Stepwise (10K Dense) | 1:8(FF) + 1:4(QKV) | **80.3** | **36.4** | **83.2** | **83.4** | 90.9 / 87.3 (89.1) | **90.3** | **91.3 / 88.3 (89.8)** | **74.7** | 90.9 | 85.2 / 85.0 (85.1) |
| MdGf-Geometric (Zero Dense) | 1:8(FF) + 1:4(QKV) | 76.8 | 20.2 | 80.5 | 80.8 | **91.3 / 87.7 (89.5)** | 85.4 | 90.3 / 87.0 (88.6) | 70.8 | 90.4 | 84.9 / 84.9 (84.9) |
| MdGf-Geometric (10K Dense) | 1:8(FF) + 1:4(QKV) | 78.9 | 27.7 | 82.4 | 82.4 | **91.3 / 87.7 (89.5)** | 88.8 | 91.0 / 88.1 (89.6) | 74.4 | **91.3** | 84.5 / 84.5 (84.5) |

Here is an itemized list of nine tasks used in the GLUE dataset, along with brief descriptions of each:

- **CoLA (Corpus of Linguistic Acceptability)**: Classify whether a given sentence is grammatically acceptable or not.

- **MNLI (Multi-Genre Natural Language Inference)**: Classify the relationship between a given premise and hypothesis as entailment, contradiction, or neutral. We use the standard test set, for which we obtained private labels from the authors, and evaluate on both the matched (in-domain) and mismatched (cross-domain) sections.

- **MRPC (Microsoft Research Paraphrase Corpus)**: Determine whether a pair of sentences express the same meaning or not.

- **QNLI (Question-answering Natural Language Inference)**: Determine whether a given question can be answered correctly using a given sentence.

- **QQP (Quora Question Pairs)**: Determine whether a pair of questions from Quora are semantically equivalent or not.

- **RTE (Recognizing Textual Entailment)**: Classify the relationship between a given premise and hypothesis as entailment or not.

- **SST-2 (Stanford Sentiment Treebank)**: Determine the sentiment of a given sentence as either positive or negative.

- **STS-B (Semantic Textual Similarity Benchmark)**: Calculate the similarity score between two sentences on a scale from 0 to 5.

These tasks cover various aspects of language understanding, including sentence acceptability, sentiment analysis, paraphrase detection, textual similarity, natural language inference, question-answering, and co-reference resolution.

Figure 5 shows the accuracy vs. fine-tuneing step curve for each of the 9 benchmarks of GLUE.

# 9 DETAILED EXPERIMENTAL SETTINGS

## 9.1 DATASETS

### 9.1.1 IMAGENET-1K

ImageNet-1K (Deng et al., 2009) is a large-scale image classification task, known as one of the most challenging image classification benchmarks. It consists of more than 1.2 million training images and

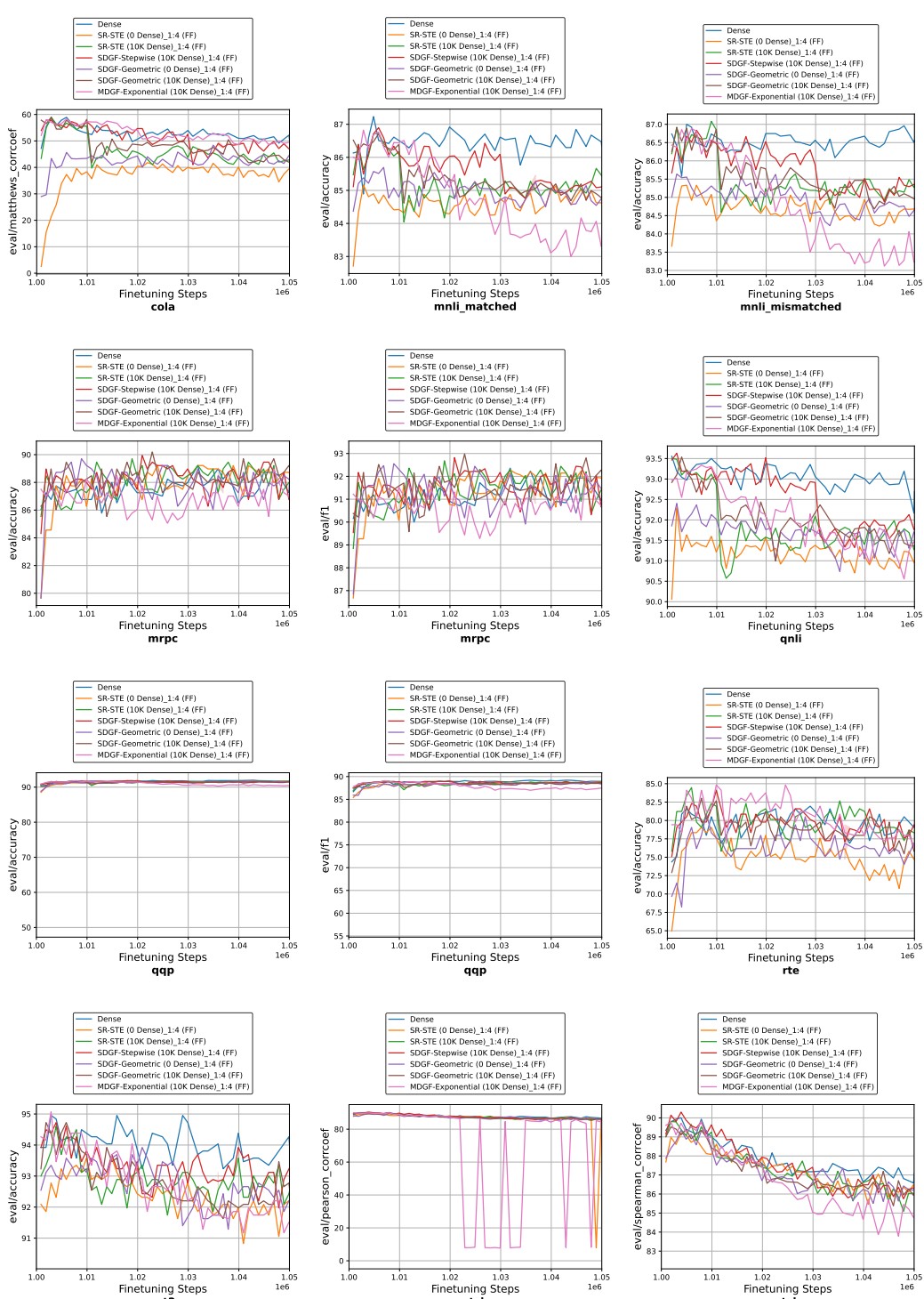

**Fig. 5: Per-task evaluations for `T5X-Base` model finetuned on the GLUE dataset for 50 K steps.**

50K validation images with a size of 224x224 pixels, each with 3 channels. Each image is labeled as one of the 1K classes. We use this dataset for studies in Section 4.1 of the main paper. For ViT and SwinV2 experiments, we use a patch size of 16. This converts the 224x224 pixel image into an input of sequence length $224/16 * 224/16 = 196$.

**Evaluation metrics.** All reported results follow standard Top-1 validation accuracy.

### 9.1.2  CIFAR10

CIFAR-10 (Krizhevsky et al., 2009) is a smaller-scale image classification dataset consisting of 10 classes. Each class has 6000 color images of 32x32 pixels in size.

**Evaluation metrics.** All reported results to follow standard Top-1 accuracy.

### 9.1.3  GLUE

The General Language Understanding Evaluation (GLUE) (Wang et al., 2019) benchmark is a collection of resources for training, evaluating, and analyzing natural language understanding systems. GLUE consists of: A benchmark of nine sentence- or sentence-pair language understanding tasks built on established existing datasets and selected to cover a diverse range of dataset sizes, text genres, and degrees of difficulty, Table 11 shows the overall score for each sparsity target using different sparsification methods.

**Evaluation metrics.** All reported results in the main paper use the overall average score.

### 9.1.4  WMT

WMT-17 (English-German) (Bojar et al., 2017) is a key benchmark in machine translation research. They hold several translation datasets across different languages. The training set consists of about 4.5 million bilingual sentence pairs from WMT 2014.

**Evaluation metrics.**  We calculate accuracy by comparing the translated output to the correct translation in the validation datasets.

### 9.2   Hyperparmeters for Different Models

#### 9.2.1   Image Classification → Vision Transformers (ViT)

We train the ViT-Base model on ImageNet-1k with hyperparameters presented in Table 12. We follow the hyperparameter setting in (Wightman, 2019) for all ViT experiments. For `ViT-Base`, we use fixed-size patches (resolution $16 \times 16$) on images with resolution 224. We also use the same hyperparameters to train ViT-Tiny model ( 3 layers, 3 attention head per layer, Embedding dimension: 192) on CIFAR-10 for initial experiments in Section 3.2 for analysing the trends of weights, gradients and optimizer moments and comparing those with SR-STE.

**Table 12: Hyperparameters used for training ViT on ImageNet-1K.**

| | |
|---|---|
| Batch Size | 256 |
| Training Epoches | 350 |
| Learning Rate | 1e-3 |
| LR Warmup Epoches | 15 |
| LR Decay schedular | Cosine |
| Decay Rate | 0.1 |
| Decay Epoches | 100 |
| Optimizer | AdamW |
| Optimizer coefs | beta1 = 0.9, beta2 = 0.999 |

The detailed list of all hyperparameters can be found at hyperparaters.yaml. For ViT-Base, the training phase takes $\approx$ 44 hours on 16 - A100 GPUs.

Figure 6 shows the Top-1 and Top-5 accuracy trends for training ViT to various sparsity targets with different sparsification techniques. We observe generally, MDGF and SDGF are better than SR-STE, especially for high-sparsity targets.

### 9.2.2 IMAGE CLASSIFICATION → SWIN TRANSFORMER V2 (SwinV2)

We train the SwinV2-Base model on imagenet-1k with hyperparameters presented in Table 13. We follow the hyperparameter setting in (Liu et al., 2022) for all SwinV2 experiments. In SwinV2-Base, we employ window sizes of 8×8 on images with resolution 256.

**Table 13: Hyperparameters used for training SwinV2 on ImageNet-1K.**

| | |
|---|---|
| Batch Size | 128 |
| Training Epoches | 350 |
| Learning Rate | 1e-3 |
| LR Warmup Epoches | 20 |
| LR Decay schedular | Cosine |
| Decay Rate | 0.1 |
| Decay Epoches | 30 |
| Optimizer | AdamW |
| Optimizer coefs | beta1 = 0.9, beta2 = 0.999 |

The detailed model configuration is the same as present in the original Microsoft research GitHub repo, SwinV2-base.yaml The detailed list of all hyperparameters was taken from config.yaml. For SwinV2-Base, the training phase takes ≈ 54 hours on 16 - A100 GPUs.

### 9.2.3 LANGUAGE UNDERSTANDING → T5X

We train the T5X-Base model on GLUE with hyperparameters presented in Table 14. We follow the hyperparameter setting in (Raffel et al., 2019) for all T5X training experiments.

The detailed model configuration is the same as present in the original Google research GitHub repo, T5X model T5X-Base's training phase takes ≈ 22 hours on 8×Google Cloud TPUv3 cores.

**Table 14: Hyperparameters used for training T5X on GLUE.**

| | |
|---|---|
| Batch Size | 128 |
| Training Steps | 100k |
| Learning Rate | 1e-3 |
| LR Warmup Steps | 1000 |
| LR Decay schedular | Constant |
| Optimizer | AdamW |
| Optimizer coefs | beta1 = 0.9, beta2 = 0.999 |

### 9.2.4 LANGUAGE TRANSLATION MODEL → Enc-Dec

We train an encoder-decoder-based model on WMT-17 with hyperparameters presented in Table 15. The model is inspired by the attention paper (Vaswani et al., 2017). We follow the hyperparameter setting in (Devlin et al., 2019) to train all models. The training phase takes ≈ 8 hours on 32 - Google Cloud TPU v3 cores.

## 10 ADDITIONAL RESULTS ON THE IMPACT OF SPARSIFICATION METHODS ON GRADIENTS

Figure 7 shows the gradient variance for all feed-forward blocks in all ViT-tiny layers. As explained in Section 4.2, lower gradient noise earlier in training would help bring down the optimiser to a higher accuracy stage early. Comparing gradient variance for MDGF-exponential and SR-STE, we see that, for MDGF-exponential the variance drops very early compared to SR-STE, hence reaching the higher final accuracy.

**Table 15: Model configurations and hyperparameters for training model on WMT.**

| | |
|---|---|
| Number of Encoder Layers | 6 |
| Number of Decoder Layer | 6 |
| Hidden Dimension Size | 1024 |
| Feed-Forward Dimension Size | 4096 |
| Number of Attention Heads | 16 |
| Max Sequence Length | 256 |
| Training Dataset | WMT-17 |
| Testing Dataset | WMT-14 |
| Batch Size | 512 |
| Training Steps | 200K |
| Learning Rate | 0.0625 |
| LR Warmup Steps | 1000 |
| Decay Factor | 0.5 |
| Optimizer | Adam |
| Optimizer coefs | beta1 = 0.9, beta2 = 0.92 |

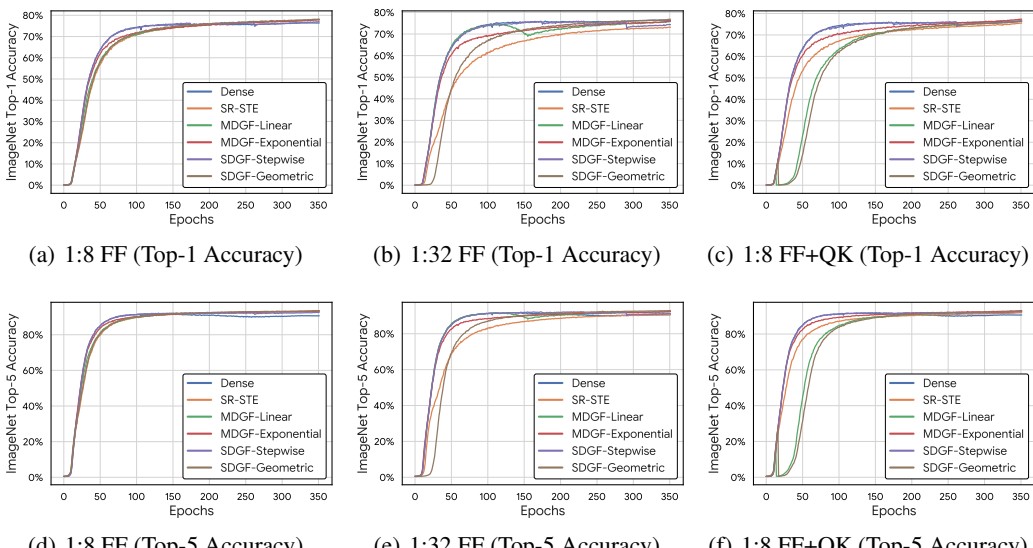

(a) 1:8 FF (Top-1 Accuracy)  (b) 1:32 FF (Top-1 Accuracy)  (c) 1:8 FF+QK (Top-1 Accuracy)

(d) 1:8 FF (Top-5 Accuracy)  (e) 1:32 FF (Top-5 Accuracy)  (f) 1:8 FF+QK (Top-5 Accuracy)

**Fig. 6: Training Epochs vs Accuracy graph for different sparsity targets. We train `ViT-Base` on ImageNet-1K.**

## 11 PARAMETERS AND FLOPS CALCULATION

We show the details of the model params, training FLOPS and Inference FLOPS calculation for Table 4.

The calculations are for ViT-Base Model (Table 16). Table 17 shows how the model parameters are being calculated for different sparsity levels. Table 18 shows the FLOPS calculation for inference.

**Table 16: ViT model parameters.**

| | |
|---|---|
| Number of Encoder Layers | 12 |
| Hidden Dimension Size | 768 |
| Feed-Forward Dimension Size | 4*768 = 3072 |
| Number of Attention Heads | 12 |
| Sequence Length | 196 [5] |

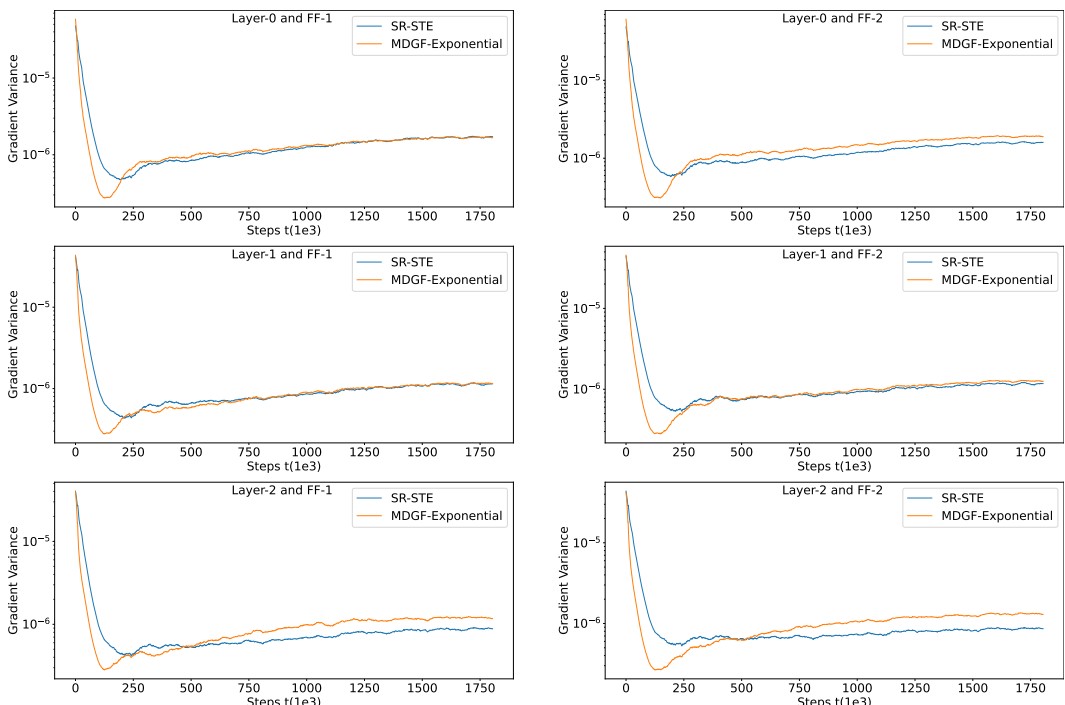

**Fig. 7: The variance of gradient w.r.t. training steps for all layers and FFs of tiny-Vit.**

**Table 17: ViT Weights Calculation.**

| Layer | Calculation | Weights |
|---|---|---|
| $W_{Query}$ | $12*768*768*\frac{N_q}{M_q}$ | $7077888*\frac{N_q}{M_q}$ |
| $W_{Key}$ | $12*768*768*\frac{N_k}{M_k}$ | $7077888*\frac{N_k}{M_k}$ |
| $W_{Values}$ | $12*768*768$ | $7077888$ |
| $W_{Output}$ | $12*768*768$ | $7077888$ |
| $W_{FF}$ | $12*2*768*3072*\frac{N_{FF}}{M_{FF}}$ | $56623104*\frac{N_{FF}}{M_{FF}}$ |
| $W_{FC}$ | $768*1000$ | $768000$ |
| Total Weights | $7077888*(2+\frac{N_q}{M_q}+\frac{N_k}{M_k}) + 56623104*\frac{N_{FF}}{M_{FF}} +768000$ | |

**Table 18: ViT Inference Flops Calculation.**

| Layer | FLOPS Calculation | Weights |
|---|---|---|
| $W_{Query}$ | $12*2*196*768*768*\frac{N_q}{M_q}$ | $2,774,532,096*\frac{N_q}{M_q}$ |
| $W_{Key}$ | $12*2*196*768*768*\frac{N_k}{M_k}$ | $2,774,532,096*\frac{N_k}{M_k}$ |
| $W_{Values}$ | $12*2*196*768*768$ | $2,774,532,096$ |
| $W_{Output}$ | $12*2*196*768*768$ | $2,774,532,096$ |
| $W_{FF}$ | $12*2*196*2*768*3072*\frac{N_{FF}}{M_{FF}}$ | $11,098,128,384*\frac{N_{FF}}{M_{FF}}$ |
| Total Weights | $2,774,532,096*(2+\frac{N_q}{M_q}+\frac{N_k}{M_k}) + 11,098,128,384*\frac{N_{FF}}{M_{FF}}$ | |

## 12 CODEBASE

Our ViT and SWINV2 codebase is made by modifying the TIMM code base of hugging-face vision transformers (Wightman, 2019). We add sparsity layers to various models and modify the training loop to support training recipes presented in this work. Similarly, we modify the jax-based codebases

for T5X and Language translation model experiments. Our anonymous codebase with training recipes can be found at https://anonymous.4open.science/r/n_m_decay_1605-E77F/.

