# OpenReview forum: "Sparsify the Weights but Let the Gradients Flow!"
_ICLR.cc/2024/Conference — ICLR 2024 Conference Withdrawn Submission_

### Official Review · Reviewer_pctS · 2023-11-01

**Soundness:** 3 good
**Presentation:** 3 good
**Contribution:** 3 good
**Rating:** 5
**Confidence:** 4

**Summary:**

This work proposes efficient approaches for improving N:M structured sparsity in DNNs by optimizing the training process. It focuses on improving the high-sparsity regions by introducing two training methods: MDGF and SDGF. These methods, utilizing decay mechanisms to manage gradient flow, overcome the challenges posed by conventional training approaches, achieving enhanced accuracy across various attention-based models in both vision and language tasks.

**Strengths:**

- The approach seems to be effective in reducing training variance and improving the model
accuracy in high-sparsity regions.
- The proposed technique is evaluated on a wide range of networks and datasets.
- The paper is well-written and easy to follow.

**Weaknesses:**

- In Table 7 of the evaluation section, the proposed method is compared to multiple baselines,
but only on, 1:32. How does it perform compared to other techniques for other sparsity
targets?
- The proposed technique seems to be mainly targeting a single baseline, the training gradient
variance (in section 4) is only compared to SR-STE. Does this issue also exist in other
techniques?
- Figures 4 a and b seem to use different axis label font styles.

**Questions:**

- There are four variances of gradient weight decay, is there an intuitive guide for which one to
select given the sparsity target or it would be trial and error?
- How do the baseline and the proposed method perform with different hyperparameters or
regularizations? Would it improve the variance of the baselines?
- For ViT-base model, it seems the proposed technique is improving the high sparsity
performance, but with a small penalty in the low sparsity area (Table 1, rows 1 and 2). Can you
explain if that is the case?
- Is it possible to apply the NM sparsity in the attention map for further acceleration?
- Given various types of operations in ViT, this paper lacks a figure to illustrate the detailed NM sparsity patterns over each types of layer (for example, will there be any correlation between these layer when applying NM sparsity over them)?

---

### Official Review · Reviewer_Q2rM · 2023-11-01

**Soundness:** 2 fair
**Presentation:** 3 good
**Contribution:** 2 fair
**Rating:** 5
**Confidence:** 3

**Summary:**

This work utilizes two gradient flow methods Mask Decay and Structure Decay for n:m structured sparsity and attain higher network accuracy than SR-STE.

**Strengths:**

1. This work is the first to exploit n:m structured sparsity under a high sparsity ratio (mainly 1:16).
2. This work details the effectiveness of two sparsity reduction methods, Mask Decay and Structure Decay, during the training phase.

**Weaknesses:**

1. I think that the gradient flow sparse training method is irrelevant to the n:m structured sparsity in this work, so the experiment results should be compared with the works of both parts separately.
Gradually decreasing the sparsity ratio during the training phase is not something new. Please compare your results to some sparse training work such as "Effective model sparsification by scheduled grow-and-prune methods."

2. If you are focusing on the performance of n:m structured sparsity, please compare your results with newer works in more dimensions rather than only focusing on SR-STE which is not designed for transformer-based DNN models:

(IDP) An algorithm–hardware co-optimized framework for accelerating n:m sparse transformers

STEP: Learning N:M Structured Sparsity Masks from Scratch with Precondition

Dynamic N:M Fine-grained Structured Sparse Attention Mechanism

Channel Permutations for N: M Sparsity

**Questions:**

Another question is why not combine both methods together? It will certainly enlarge the search space for better results. Hope to see an analysis of the two gradient flow methods in a theoretical aspect.

---

### Official Review · Reviewer_mQGH · 2023-11-02

**Soundness:** 2 fair
**Presentation:** 2 fair
**Contribution:** 2 fair
**Rating:** 3
**Confidence:** 4

**Summary:**

The paper identifies increased variance in gradients as a factor which negatively impacts the quality of models pruned with n:m sparsity. To remedy this, two sparse training techniques, mask and structure decay gradient flow (MDGF and SDGF) are proposed, which gradually decay the magnitude of sparsity masks. These successfully reduce gradient variance. Experiments are conducted on ViT, Swin, and ResNet models on ImageNet, T5 on GLUE, and an encoder/decoder transformer on English-German translation, and show MDGF or SDGF a SR-STE baseline.

**Strengths:**

1. The paper presents relatively simple set of methods for pruning which nevertheless seem to consistently improve performance over the baselines. There is a broad discussion of existing prior approaches to pruning.
2. The methods are evaluated on a wide array of benchmarks, and the appendix includes ablation studies.

**Weaknesses:**

1. The paper's clarity could be improved. I found it hard to follow and the methods proposed to be incompletely specified. Improved figures and a full algorithm specification could be helpful here. At a higher level, the paper's structure is odd: It first proposes new methods (MDGF/SDGF) and then shows they introduce different training dynamics (reducing gradient variance). To me it seems this should be reversed: First identify the problem with existing methods (too much gradient variance), then develop a method that solves this problem. This argument could be strengthened by showing what the variance of SR-STE is relative to training without any pruning.
2. The experiments do not include any measure of variance across training runs. Given the stochasticity of training and pruning, this is critical to determining how well the methods perform relative to each other, especially given that the differences in performance are relatively small.
3. It is not clear to me how relevant the discussion of training performance in Section 5.1 is. While sparsity will indeed reduce parameter (and hence gradient and optimizer state) counts and can reduce the number of flops, I do not think this translates so directly to improved performance: Data movement overheads are often paramount and not necessarily linear in the number of parameters; further, there are practical overheads (e.g., data marshalling to actually run the n:m format) which complicate this. A more nuanced discussion would serve better.
4. Related to (3), I am confused about what exactly Table 4 is showing. The caption indicates "training recipe performance", as does the paragraph heading ("Training performance"), yet the paragraph refers to "_inference_ cost measured in FLOPs" (emphasis added).

**Questions:**

1. Please improve the clarity of the paper; see above for some suggestions. The discussion in Section 4 would also benefit from more precisely defining the metrics being shown and the study done.
2. In Section 4, what is the gradient variance for training a model with no pruning?
3. Please add variance to the accuracy results in Section 5. If this is done and continues to show the trends indicated, I am willing to raise my score.
4. Please clarify the discussion of training performance in Section 5.1 (see above).